# Astatine-211-Labeled Therapy Targeting Amino Acid Transporters: Overcoming Drug Resistance in Non-Small Cell Lung Cancer

**DOI:** 10.3390/ijms262110736

**Published:** 2025-11-05

**Authors:** Sifan Feng, Kentaro Hisada, Haruna Yorifuji, Yoshifumi Shirakami, Kazuko Kaneda-Nakashima

**Affiliations:** 1Radiation Biological Chemistry, Department of Chemistry, Graduate School of Science, The University of Osaka, Osaka 560-0043, Japan; 2Institute for Radiation Sciences, The University of Osaka, Osaka 565-0871, Japan; yoshifumi_shirakami@irs.osaka-u.ac.jp; 3Core for Medicine and Science Collaborative Research and Education, Forefront Research Center, Graduate School of Science, The University of Osaka, Osaka 560-0043, Japan

**Keywords:** non-small cell lung cancer, drug resistance, amino acid transporters (LAT1, ASCT2, xCT), metabolic reprogramming, Astatine-211, targeted α-particle therapy, radiopharmaceuticals

## Abstract

Non-small cell lung cancer (NSCLC) remains a leading cause of cancer mortality, with therapeutic resistance posing the primary barrier to durable outcomes. Beyond genetic and epigenetic alterations, amino acid transporter-driven metabolic reprogramming—mediated by LAT1 (SLC7A5), ASCT2 (SLC1A5), and xCT (SLC7A11)—supports tumor proliferation, redox homeostasis, and immune escape. Their preferential expression in NSCLC highlights their potential as therapeutic targets and predictive biomarkers. In parallel, α-particle therapy has gained attention for its capacity to eradicate resistant clones through densely clustered, irreparable DNA double-strand breaks. Astatine-211 (^211^At) combines a clinically relevant half-life, high linear energy transfer, and predictable decay scheme, positioning it as a unique candidate among α-emitters. Preclinical studies of ^211^At-labeled transporter ligands, particularly LAT1-targeted conjugates, demonstrate potent tumor suppression and synergy with targeted therapy, chemotherapy, radiotherapy, immunotherapy, and ferroptosis inducers. Advances in radiochemistry, delivery systems (antibodies, peptides, and nanocarriers), and PET tracers such as [^18^F]FAMT and [^18^F]FSPG collectively support a theranostic framework for patient stratification and adaptive dosing. By linking transporter biology with α-particle delivery, ^211^At-based theranostics offer a mechanistically orthogonal strategy to overcome resistance and heterogeneity in NSCLC. Successful translation will depend on precise dosimetry, scaffold stabilization, and biomarker-guided trial design, enabling progression toward first-in-human studies and future integration into multimodal NSCLC therapy.

## 1. Introduction

Lung cancer remains the leading cause of cancer-related mortality worldwide [1]. Non-small cell lung cancer (NSCLC) accounts for ~85% of cases [2] and continues to show limited long-term survival despite advances in surgery, chemotherapy, radiotherapy, targeted agents, and immunotherapy [3]. Therapeutic resistance—driven by both genetic and non-genetic mechanisms—remains the primary barrier to achieving durable benefits [4,5]. Among these mechanisms, metabolic reprogramming is a hallmark of adaptation under therapeutic pressure [6]. Amino acid transporters such as LAT1 (SLC7A5) and ASCT2 (SLC1A5) sustain nutrient influx, engage proliferative signaling, and contribute to resistance [7,8], while exhibiting preferential tumor expression that renders them actionable targets; however, no LAT1- or ASCT2-directed agents have progressed to late-phase trials [9,10,11].

Targeted α-particle therapy deposits large amounts of energy within a very short path length, generating densely ionizing tracks that induce complex and primarily irreparable DNA double-strand breaks, while maintaining cytotoxicity largely independent of oxygen tension [12]. Astatine-211 (^211^At) exhibits favorable radiophysical and radiochemical characteristics—including high linear energy transfer, short tissue penetration range, and a simple decay scheme yielding one α-particle and ending in stable ^207^Pb—that collectively support reliable radiopharmaceutical design and centralized clinical distribution [13]. Owing to these features, ^211^At is regarded as a particularly promising radionuclide for clinical translation. While prior reviews have summarized NSCLC resistance mechanisms [4,14] or radionuclide therapies as independent topics [14,15], no review has comprehensively integrated amino acid transporter biology with ^211^At-based α-particle therapeutics. This review seeks to bridge this gap and propose a translational framework for overcoming therapeutic resistance in NSCLC.

## 2. Therapeutic Approaches and Mechanisms of Drug Resistance in NSCLC

### 2.1. Therapeutic Approaches

Management of NSCLC is increasingly biomarker-driven [16] and differs substantially across disease stages [17]. Surgical resection remains the only curative modality for stage I–II disease and for selected stage IIIA patients, typically combined with perioperative chemotherapy or immunotherapy [18]. For medically inoperable early-stage NSCLC, stereotactic body radiotherapy (SBRT) provides a practical, non-invasive alternative with excellent local control rates [19,20]. In unresectable stage III disease, concurrent chemoradiation followed by consolidation durvalumab represents the current standard of care for eligible patients [21]. In the metastatic setting, first-line therapy is guided by biomarkers. Patients with oncogenic drivers such as EGFR, ALK, ROS1, or KRASG12C derive substantial benefit from targeted agents [22]. By contrast, immune checkpoint inhibitors (ICIs), either alone or combined with platinum-based chemotherapy, are recommended in tumors without actionable oncogenic drivers [23]. PD-L1 expression serves as a predictive biomarker of response to ICI monotherapy, particularly in tumors with high PD-L1 expression [16,24]. However, in oncogene-driven NSCLC, ICIs generally provide limited benefit [25,26]. Overall, although targeted therapy and immunotherapy have reshaped the therapeutic landscape, durable disease control and long-term survival remain unmet needs for many patients [27].

### 2.2. Resistance Mechanisms

Therapeutic resistance in NSCLC is highly heterogeneous [28] and often involves multiple concurrent processes [29], with strong context dependence by inhibitor class, tumor genotype, and residual disease status [30]. Broadly, mechanisms span genetic, epigenetic, metabolic, and microenvironmental processes [31,32,33]. Representative resistance mechanisms—including on-target/bypass mutations, epigenetic remodeling, EMT/stemness, drug efflux and metabolism, immunosuppressive TME, metabolic adaptation, DNA damage response (DDR) rewiring, and tumor persistence/dormancy—are summarized in Table 1 and schematically illustrated in Figure 1.

### 2.3. Modality-Specific Resistance Examples

For tyrosine kinase inhibitors, on-target mutations such as EGFR T790M/C797S [73] and bypass activation of MET–ERBB3–PI3K/AKT are prototypical routes that compromise drug binding or reactivate downstream signaling [36]. For immune checkpoint inhibitors, resistance commonly involves defective antigen presentation (e.g., beta-2 microglobulin (B2M) or human leukocyte antigen class I (HLA-I) loss [74]) and T-cell exclusion/exhaustion driven by suppressive myeloid cells and Tregs [75]; adaptive upregulation of alternative checkpoints (TIM-3, LAG-3, TIGIT, VISTA) in non-inflamed or metabolically reprogrammed TMEs further limits benefit [75,76]. For chemotherapy and radiotherapy, DNA-damage response activation [66] and hypoxia-induced HIF-1 signaling are dominant drivers [77]. Other adaptations—including small cell transformation [78] and EMT [79], epigenetically sustained drug-tolerant persisters [69,80]—also contribute to treatment failure.

### 2.4. Shared Adaptive Programs Across Modalities

Although resistance manifests differently across therapeutic modalities, several convergent adaptive programs emerge as key drivers of treatment failure. These shared adaptations include (i) metabolic adaptation, such as enhanced glutamine and cystine flux via LAT1, ASCT2, and xCT, which sustain redox balance, biosynthesis, and immune evasion; (ii) epigenetic plasticity that maintains reversible drug-tolerant states; (iii) alterations in drug transport and metabolism involving ABC transporters and CYP/UGT enzymes, which modify intratumoral drug exposure; and (iv) persistence or dormancy programs that facilitate relapse from minimal residual disease. Collectively, these adaptations underlie the limited durability of current treatments. Biomarker-guided next-generation inhibitors, rational therapeutic combinations, and optimized sequencing can overcome specific resistance routes, such as MET-amplified resistance to EGFR TKIs. Nevertheless, tumors frequently retain shared metabolic vulnerabilities [81]. In particular, metabolic plasticity mediated by amino acid transporters such as LAT1, ASCT2, and xCT constitutes a tractable vulnerability across various treatment contexts, as it supports redox control, biosynthesis, and immune evasion [60,82]. This adaptation directly links resistance biology to therapeutic metabolism and to radiolabeled theranostic strategies.

## 3. Amino Acid Transporters in NSCLC: Structure, Function, and Therapeutic Targeting

Beyond genetic and epigenetic mechanisms, therapeutic resistance in NSCLC is also sustained by metabolic adaptations. Among these, amino acid transporters play a crucial role in sustaining tumor growth and maintaining redox balance, while also contributing to immune evasion and treatment resistance, thereby positioning them as critical mediators between tumor metabolism and therapeutic outcomes.

### 3.1. Overview and Structural Features

Amino acid transporters of the SLC superfamily orchestrate cellular nutrient flux to meet the anabolic and redox demands of cancer cells, thereby representing potentially tractable therapeutic targets in oncology [83]. In NSCLC, three transporters—LAT1 (SLC7A5), ASCT2 (SLC1A5), and xCT (SLC7A11)—have been most extensively studied for their roles in nutrient uptake, mTORC1 activation, and redox control. LAT1 is a nonglycosylated light chain that forms a disulfide-linked heterodimer with the glycosylated heavy chain 4F2hc (SLC3A2), which mediates Na-independent uptake of large neutral amino acids such as leucine and phenylalanine [84]. ASCT2 functions as a Na^+^-dependent exchanger with a substrate preference for glutamine. N-linked glycosylation at Asn163 and Asn212 ensures its membrane stability [85]. This transporter supports the “glutamine addiction” phenotype of NSCLC and is commonly overexpressed in tumors compared with adjacent lung tissue [85,86]. xCT, the light chain of system X_c_^−^, heterodimerizes with 4F2hc to mediate equimolar cystine–glutamate exchange [87,88]. Imported cystine is reduced intracellularly to cysteine, the rate-limiting precursor of glutathione, thereby sustaining redox homeostasis and therapy resistance [88]. The stability of xCT is reinforced by CD44 variant isoforms [89] and NRF2-dependent transcription under oxidative stress [90]. High SLC7A11 expression correlates with poor prognosis [91] and synthetic lethality in KRAS-mutant lung adenocarcinoma models [92]. Collectively, LAT1, ASCT2, and xCT constitute a metabolic network in which glutamine efflux via ASCT2 fuels LAT1-mediated leucine uptake to activate mTORC1. In contrast, xCT-mediated cystine uptake maintains glutathione-based antioxidant defense, underscoring these transporters as central metabolic nodes in NSCLC. The key structural and functional features of these transporters are summarized in Table 2.

### 3.2. Roles in Tumor Metabolism and the Microenvironment

These transporters not only ensure nutrient availability but also coordinate oncogenic signaling with stress adaptation in NSCLC. Leucine uptake via the LAT1 4F2hc complex plays a central role in mTORC1 activation. Binding of leucine to Sestrin2 disrupts its inhibitory interaction with GATOR2 [102], thereby enabling Rag GTPase-dependent lysosomal recruitment of mTORC1 [103]. Glutaminolysis further provides α-ketoglutarate, which reinforces Rag-mediated activation [104]. In colorectal cancer models, oncogenic KRAS amplifies this circuit by upregulating LAT1, ASCT2, and SNAT2 [105]. LAT1 inhibition with BCH reduces mTOR phosphorylation and synergizes with gefitinib to reduce NSCLC cell viability [106]. ASCT2 mediates glutamine import, which is essential for anaplerosis, maintaining redox balance, and promoting proliferation. Imported glutamine can feed exchange cycles that support leucine influx and sustain mTORC1 activation [107]. In the tumor microenvironment (TME), glutamine depletion restricts T-cell proliferation and cytokine production [108,109]. Pharmacologic inhibition with V-9302 synergizes with EGFR-TKIs [97], promoting apoptosis, suppressing autophagy, and enhancing the activity of CD8^+^ T cells [110]. xCT maintains glutathione biosynthesis by exchanging extracellular cystine for intracellular glutamate [87,88]. This glutathione pool buffers therapy-induced oxidative stress [111]. KEAP1/NRF2 mutations drive constitutive SLC7A11 expression [112], while the induction of ATF4 further enhances stress responses [60]. Blockade of xCT with sulfasalazine depletes GSH, radiosensitizes tumors [113], and augments cisplatin efficacy [114]. At the downstream node, GPX4 detoxifies lipid peroxides, thereby suppressing ferroptosis [115]. In NSCLC models, perturbing redox–ferroptosis defenses increases ferroptosis susceptibility (with links to GPX4 downregulation) [116], and, more broadly across drug-resistant solid tumors, inhibition of the System X_c_^−^/GSH/GPX4 axis restores ferroptosis sensitivity [117].

### 3.3. Immune and Expression Profiles in NSCLC

Beyond intrinsic metabolism, amino acid transporters play critical roles in modulating antitumor immunity. Tumor-derived kynurenine is imported into T cells via the LAT1 (SLC7A5) transporter, thereby activating AHR signaling [118] to drive regulatory T-cell differentiation [119]. Effector T cells depend on LAT1 for clonal expansion [120], whereas Tregs compensate through methionine uptake via SLC43A2 [121]. Under metabolic stress, glutamine deprivation can induce PD-L1 via EGFR–ERK–c-Jun signaling (shown in renal cancer models), suggesting a metabolism–immune checkpoint link that may extend to other contexts [122], thereby suppressing antitumor immunity. IDO1 expression within tertiary lymphoid structures correlates with Treg infiltration [123]. In resected lung adenocarcinomas, IDO1 upregulation is associated with reduced T-cell density [124]. Collectively, LAT1, ASCT2, and xCT coordinate metabolic and immune adaptations that promote tumor growth and immune evasion. These findings underscore the dual role of amino acid transporters as therapeutic targets and biomarkers of immunotherapy response.

In parallel with their immunomodulatory functions, transporter expression in NSCLC is subtype-specific and carries prognostic significance. LAT1 is overexpressed in squamous cell carcinoma (91%) and large-cell carcinoma (67%), but is less frequent in adenocarcinoma (29%), and is absent in normal bronchial and alveolar epithelium [125]. ASCT2 is expressed in ~66% of tumors and correlates with advanced stage, lymphovascular invasion, and poor prognosis, particularly in adenocarcinoma [98]. Approximately 12% of adenocarcinomas coexpress LAT1 and ASCT2, a profile associated with a poor outcome [126]. In NSCLC, xCT is frequently overexpressed, promoting tumor progression [101]. Across cancers, elevated SLC7A11 maintains redox balance and suppresses ferroptosis [60] and has been linked to poor survival in glioma [127]. Clinically, LAT1 positivity predicts worse overall survival (5-year OS 51.8% vs. 87.8%) [125].

Mechanistically, oncogenic and microenvironmental cues further reinforce transporter activity [128]. Hypoxia drives HIF-2α-dependent regulation of LAT1 [129], NRF2 binding at the SLC7A11 promoter augments cystine uptake [130], and EZH2-dependent H3K27 trimethylation shapes amino acid metabolic programs, implicating the LAT1–mTORC1 axis [131]. ASCT2 requires N-linked glycosylation for membrane stability [132], whereas LAT1 transport activity is dependent on membrane cholesterol [133]. Functionally, glutamine imported via ASCT2 can be exchanged for leucine via LAT1 to activate mTORC1 [134], while xCT-mediated cystine uptake sustains glutathione synthesis [100,134]. KEAP1/NRF2-driven xCT activity promotes a resistance phenotype, as measured by [^18^F]FSPG PET [112]. Moreover, KEAP1 knockout increases resistance to KRAS G12C inhibitors [135], highlighting transporter-linked therapeutic adaptation.

Overall, LAT1, ASCT2, and xCT are overexpressed through oncogenic, hypoxic, and epigenetic programs, support metabolic and redox demands, and predict poor outcomes, thereby establishing them as both biomarkers and therapeutic targets in NSCLC. Their selective overexpression and cell-surface accessibility further create molecular gateways for precision delivery. Transporter-directed ligands can be engineered to carry α-emitters such as ^211^At, thereby linking metabolic targeting to radiotheranostic strategies and bridging fundamental transporter biology with the radiochemical and translational frameworks discussed in the following sections.

## 4. Physical and Radiobiological Basis of ^211^At Therapy

### 4.1. Physical & Radiobiological Properties

^211^At is produced via the ^209^Bi(α,2n)^211^At nuclear reaction on bismuth targets using α particles of approximately 28–29 MeV. Its ~7.2 h half-life accommodates synthesis, quality control, and distribution while limiting prolonged exposure [13]. After 48 h, less than 1% of the initial activity remains [136]. During its decay, ^211^At yields two α-particle energies of approximately 5.87 and 7.45 MeV [137]. α-particles deliver high linear energy transfer (LET, ~50–230 keV/μm) and induce densely clustered DNA double-strand breaks that are difficult to repair [12]. In soft tissue, their track range is ~50–80 μm (~5–8 cell diameters), thereby restricting cytotoxicity to targeted cells [138]. The oxygen-enhancement ratio is ~1.0 [139], indicating relative hypoxia independence compared with photons or β-particles.

### 4.2. Comparison with Other Therapeutic Radionuclides

Among α-emitters, actinium-225 (^225^Ac; t_1_/_2_ = 9.92 d) yields five α emissions but suffers from daughter redistribution and dosimetry complexity [140,141]. In contrast, bismuth-213 (^213^Bi; t_1_/_2_ ≈ 45.6 min) requires rapid pharmacokinetics and on-site preparation, thereby limiting broader applicability [142]. By comparison, β-emitters such as iodine-131 (^131^I; t_1_/_2_ = 8.05 d) provide long-range cross-fire but at the cost of higher collateral exposure [143,144]. Lutetium-177 (^177^Lu; t_1_/_2_ = 6.65 d) offers medium-range β emission with favorable imaging capabilities [145,146]. ^211^At uniquely generates short-range α tracks (~50–80 μm) capable of eradicating micrometastases and minimal residual disease [147], although effective delivery requires homogeneous target expression and efficient tumor penetration [148,149]. Clinical proof of concept for systemic α-therapy has been established by radium-223 in mCRPC [150], positioning ^211^At as a promising candidate due to its balanced half-life, potency, and logistics.

### 4.3. Radiolabeling Strategies of ^211^At

The weak C–At bond (~49 kcal·mol^−1^) and in vivo deactivation necessitate the development of robust labeling strategies [151]. Electrophilic astatination of aryl stannanes remains the gold standard [13], whereas boronic acid derivatives provide tin-free alternatives [152]. Diaryliodonium salts exhibit high regioselectivity and functional group tolerance [153]. Controlling reaction conditions—using mild temperature and short reaction times—has been reported to mitigate radiolysis and improve stability [151]. Several design principles can be distilled: (i) employ mild, rapid electrophilic conditions compatible with the 7.2 h half-life; (ii) favor para (and, to a lesser extent, ortho) aromatic substitution to improve labeling performance and stability [13]; and (iii) leverage the side-chain tolerance observed for LAT1-privileged amino acid mimetics, allowing incorporation of para-aryl astatine or small prosthetic groups without abolishing transport [13,154]. These guidelines explain why para-aryl astatination of LAT1-privileged amino acid mimetics consistently yields high tumor-to-blood ratios [154,155].

### 4.4. Targeting Amino Acid Transporters with ^211^At

The central rationale is to integrate the unique radiophysical properties of ^211^At with the metabolic dependencies of NSCLC. A schematic overview of this therapeutic concept is provided in Figure 2, which illustrates the workflow of ^211^At-labeled LAT1-targeted therapy, including target identification, vector design (antibody/peptide/small molecule), radiolabeling, tumor accumulation, and the induction of clustered DNA double-strand breaks by short-range, high-LET α-particles.

LAT1 (SLC7A5) is widely regarded as a leading amino acid transporter target, whereas ASCT2 (SLC1A5) and xCT (SLC7A11) are often discussed as complementary nodes that help support mTORC1 activity and redox balance. From a chemical standpoint, aryl C–At linkages obtained via electrophilic astatination—using, for example, destannylation, astatodeboronation, or diaryliodonium precursors—represent established labeling approaches, and are compatible with amino acid–mimetic scaffolds and late-stage modifications.

Two LAT1-targeted chemotypes currently define the ^211^At radiopharmaceutical landscape. 2-[^211^At]AAMP, a phenylalanine analogue, exhibits LAT1-dependent uptake, α-radiation-induced cytotoxicity, and survival benefit in tumor-bearing models [155]. In parallel, ^211^At-AAMT, an α-methyl-L-tyrosine derivative, shows dose-dependent activity, ranging from growth suppression at 0.4 MBq/mouse in PANC-1 xenografts to complete inhibition of B16F10 lung metastases at 1 MBq/mouse [63]. Structural studies indicate that LAT1 favors bulky neutral amino acids and tolerates para-substituted aromatic rings and α-methyl handles without loss of recognition [156]. Efficient labeling under mild electrophilic conditions [151] is required to match the ~7.2 h half-life of ^211^At [13]. Pharmacologic modulation of renal clearance offers an additional strategy: probenecid pretreatment prolongs circulation [157] and increases tumor uptake [154] of [^211^At]AAMP, thereby enhancing efficacy [58]. Together, these studies establish design rules—including LAT1-privileged scaffolds, para-aryl astatination, α-methyl stabilization, and circulation modulation—that consistently yield high tumor-to-blood ratios and reproducible in vivo efficacy in LAT1^+^ models.

ASCT2-mediated glutamine uptake and LAT1-facilitated essential amino acid transport are linked to mTORC1 activation [134]. LAT1 inhibitors such as JPH203 suppress amino acid signaling and global protein translation [158], although off-target effects on SNAT2 and LAT1 complicate selectivity [159]. Dual inhibition of ASCT2 and LAT1 has therefore been proposed to more effectively collapse the glutamine–leucine–mTOR axis [160].

In parallel, xCT (SLC7A11) sustains cystine uptake for glutathione biosynthesis and redox defense [100]. NSCLC can exhibit dysregulated KEAP1–NRF2 signaling, which drives xCT activation [100], and [^18^F]FSPG PET enables noninvasive readouts of NRF2-linked system X_c_^−^ activity [112]. Pharmacologic blockade of system X_c_^−^ depletes glutathione, increases lipid peroxidation, and radiosensitizes tumors in vivo [100,161,162]. In addition, erastin and sorafenib trigger ferroptosis through xCT inhibition [163,164]. Thus, ASCT2 and xCT represent rational co-targets that can be layered onto LAT1-directed α-radiopharmaceuticals, producing synthetic lethality through the combined deprivation of amino acids and the delivery of DNA damage.

### 4.5. Applications in Other Malignancies

Beyond NSCLC, amino acid transporter-targeted ^211^At radiopharmaceuticals have demonstrated potent and selective antitumor efficacy across multiple malignancies, highlighting their broad translational promise. In pancreatic cancer (PANC-1) xenografts, intravenous ^211^At-AAMT (0.4 MBq/mouse) significantly suppressed tumor growth, whereas BCH pretreatment reduced tumor uptake, confirming LAT1-mediated delivery. In the same preclinical study, a melanoma (B16F10) experimental lung-metastasis model treated with ^211^At-AAMT (0.1–1 MBq/mouse) showed markedly fewer pulmonary nodules without appreciable weight loss, indicating strong therapeutic efficacy with minimal systemic toxicity [63]. In ovarian cancer (SKOV3) xenografts, the LAT1-targeting analog 2-[^211^At]AAMP achieved selective uptake and significantly improved survival at tolerated activities [155]. Across glioma models—C6, U-87MG, and GL261—^211^At-labeled phenylalanine analogs produced dose-dependent tumor suppression at 0.1–1 MBq/mouse, with BCH competitively reducing cellular uptake, consistent with system-L/LAT1 transport [165]. In orthotopic GL261 glioblastoma, intratumoral ^211^At-anti-syndecan-1 yielded robust intracranial control and markedly prolonged survival, illustrating durable responses in brain tumors [166]. Extending beyond epithelial malignancies, bone and soft-tissue sarcomas also responded to LAT1-targeted ^211^At-AAMT-O-Me-L in preclinical studies, supporting its applicability to mesenchymal tumors [167]. Collectively, these cross-cancer findings verify mechanism-driven uptake, dose-dependent efficacy, and favorable tolerability, providing a strong biological foundation for adapting LAT1/ASCT2/xCT-targeted α-radiotherapeutics to NSCLC and guiding the translational strategies discussed below.

### 4.6. Delivery Platforms for Enhanced Retention

While small-molecule transporter ligands enable rapid tumor penetration and efficient uptake, their clinical translation is limited by their radiochemical fragility, rapid clearance, and restricted structural diversity. To address these limitations, biological and nanotechnology platforms, such as monoclonal antibodies (mAbs), antibody–drug conjugates (ADCs), engineered peptides, and nanocarriers, are being developed as complementary systems for ^211^At-based transporter-targeted therapies. These modalities extend beyond substrate analogues by enabling multivalent binding, prolonged circulation, and modular payload integration, thereby broadening the therapeutic landscape for resistant NSCLC.

#### 4.6.1. Antibodies and ADCs

CD98hc (SLC3A2), which interacts with integrins and regulates adhesion-associated signaling [168], forms a disulfide-linked heterodimer with LAT1 [169]. Anti-LAT1 antibodies bind extracellular epitopes, induce internalization, and inhibit amino acid uptake, typically leading to antibody-dependent cellular cytotoxicity (ADCC) and tumor suppression in xenograft models. These effects suppress tumor growth in xenograft models [170]. CD98hc antibodies, such as IGN523, exhibit potent ADCC and achieve tumor control comparable to carboplatin in NSCLC xenografts [171]. ADC formats extend this principle. For example, CD98hc–DM1 and CD98hc–MMAE conjugates exploit antigen-mediated endocytosis and lysosomal release, producing mitotic catastrophe or DNA damage [171,172,173,174]. Payloads such as auristatins (MMAE) disrupt microtubules [175], whereas pyrrolobenzodiazepine (PBD) dimers generate lethal DNA cross-links [176]. For antibody-based targeted delivery of cytotoxic payloads, improving the therapeutic index through target/payload/linker optimization is a central design goal [177]. At the same time, tumor penetration and payload-related toxicities remain important considerations [178].

#### 4.6.2. Peptides

Peptides offer intermediate pharmacokinetics, balancing target specificity with short circulation times [179] that are compatible with the ^211^At half-life. Cyclization, peptidomimetic engineering, and multivalency strategies further improve stability and avidity. ^211^At-labeled RGD peptides demonstrate high tumor uptake, with reduced blood radioactivity that improves tumor-over-blood contrast, and reproducible antitumor efficacy [180,181]. Although LAT1-selective peptides were not evaluated in these studies, these studies confirm that short-lived α-emitters can be effectively deployed using peptide scaffolds.

#### 4.6.3. Nanocarriers

Table 3 provides a mechanism-oriented synopsis of amino acid transporter-directed strategies and 211At conjugates in NSCLC, with representative agents and key preclinical readouts, where indicated. Early translational data are also noted.

Nanocarriers address the challenges of rapid clearance and deactivation by enhancing circulation and tumor accumulation. LAT1-targeted liposomes (100–135 nm, L-DOPA-functionalized) demonstrate transporter-mediated uptake and improved brain tumor penetration [191]. PLGA nanoparticles decorated with glutamate–polyoxyethylene stearate recycle LAT1 and enhance tumor uptake, leading to improved in vivo efficacy [192]. Glutamine-conjugated PLGA nanoparticles similarly increase tumor accumulation and enable tumor scintigraphy [193]. ^211^At can also be rapidly loaded onto gold nanoparticles (~5 min, high yield), achieving systemic antitumor activity [194]. PEGylation and tumor-penetrating peptides (e.g., iRGD) further enhance pharmacokinetics and intratumoral penetration [195,196]. Hybrid PET-theranostic nanoplatforms can aid patient selection, dosimetry, and therapy monitoring, although scalability and regulatory barriers remain [197,198].

### 4.7. Toward Clinical Translation

Across α-therapy applications beyond NSCLC, variability in response depth and durability indicates recurrent constraints relevant to ^211^At transporter-targeted strategies. Intratumoral heterogeneity of radiopharmaceutical uptake [199] and spatially uneven target/transporter expression [200] can generate micro-dosimetric cold spots, compromising tumor control. Meanwhile, activation of ATM/BRCA1-linked DNA-damage responses can mitigate radiation cytotoxicity [201] and regional hypoxia/perfusion deficits [202], further limiting the effectiveness of α-particle lethality and dose deposition. Metabolic compensation via ASCT2/xCT may restore amino acid flux when glutamine transport is perturbed [203], and pharmacokinetic liabilities—including in vivo deactivation and short half-life-related constraints—reduce target exposure [204]. Within this context, optimization of dosimetry and pharmacokinetics, structured mitigation of hematologic and thyroidal toxicity, and biomarker-guided selection using transporter PET, together with GMP-ready automation, are central considerations for advancing ^211^At in NSCLC.

#### 4.7.1. Dosimetry and Pharmacokinetics

^211^At α-tracks traverse on the order of several tens of micrometers in soft tissue [138], underscoring the importance of microdistribution and tumor cell geometry beyond organ-averaged dosimetry [205]. Bone marrow is a key organ for systemic dose considerations [205], whereas in vivo deactivation increases exposure to the thyroid and stomach. Prophylactic administration of potassium iodide or perchlorate effectively reduces thyroid and gastric uptake [206]. Preclinical studies show that hematologic depression is typically transient, with blood counts recovering by approximately 28 days after sub-myeloablative dosing [207]. Scaffold stability further influences dosimetry. Aryl–astatine conjugates can exhibit variable in vivo deastatination, particularly upon internalization [13], whereas SAGMB prosthetics are designed to improve bond integrity, with halogen-bond interactions implicated relative to classical aryl–astatine motifs [208]. Clinically, intraperitoneal administration of ^211^At-MX35 F(ab′)_2_ was well tolerated [209], supporting the safety of localized α-radioimmunotherapy. By contrast, LAT1-directed small molecules remain at the preclinical stage, with no large-animal data available [210], defining a key translational gap. Beyond intraperitoneal applications, systemic α-therapy has clinical proof-of-concept with ^223^Ra in mCRPC [150], while ^225^Ac-PSMA is under clinical evaluation [211], providing a translational precedent for ^211^At.

#### 4.7.2. Safety and Toxicity Mitigation

In systemic ^211^At therapy, normal-organ exposure is substantial and thyroid uptake of free astatine requires blocking [15]. Scaffold stabilization strategies, including neighboring-group substitution and hydrophilic linker engineering, reduce radiolysis and minimize off-target uptake [212]. Physiologically based pharmacokinetic modeling shows delivery-limited and heterogeneous tumor uptake, supporting the rationale for patient-specific dosimetry [213]. Together, these insights provide a framework for rational safety management in transporter-targeted ^211^At therapies.

#### 4.7.3. Patient Selection and Trial Design

LAT1- and xCT-targeted PET tracers are emerging as dynamic biomarkers that capture transporter activity and provide non-invasive readouts of pathway activation that may support therapy monitoring. LAT1-targeted imaging with [^18^F]FAMT has been evaluated as a functional biomarker for monitoring LAT1 activity and prognosis in NSCLC [58,214]. Likewise, [^18^F]FSPG PET non-invasively quantifies xCT activity and KEAP1/NRF2-driven metabolic adaptation [112], underscoring the potential of transporter-based imaging to inform patient selection and patient-specific dosimetry in future ^211^At trials.

Building on these functional insights, several tracers have entered clinical validation as tools for stratification. [^18^F]FAMT correlates with LAT1 expression and may aid identification of LAT1^+^ NSCLC suitable for therapy planning, pending prospective validation [215]. [^18^F]FBPA PET reflects LAT1 expression in lung and mediastinal tumors [216]. [^18^F]FSPG PET quantifies redox metabolism and NRF2 activity [112], providing a complementary approach for selecting patients most likely to benefit from transporter-targeted α-therapy.

Early-phase protocols should incorporate theranostic, patient-specific dosimetry, using isotopic surrogates or trace therapeutic administrations to individualize activity [217]. GMP-compliant automation is advancing. The COSMiC-Mini dry-distillation system produces GMP-compliant, sterile [^211^At]NaAt with complete quality control, with total production achieved within ~3 h from target setup [218]. The same platform supports the automated synthesis of [^211^At]MABG with high radiochemical purity in ~28 min [219]. Such closed, GMP-compliant platforms not only ensure radiochemical reproducibility and sterility but also meet regulatory expectations, thereby lowering the barrier for multi-center clinical trials.

Collectively, translational studies emphasize that the success of ^211^At-based transporter-targeted theranostics will hinge on precise dosimetry, scaffold stabilization, patient stratification with companion imaging, and integration with established NSCLC treatments. With optimized chemistry and biologic delivery systems, transporter-guided ^211^At therapy is poised to advance toward first-in-human evaluation.

## 5. Integration with Existing NSCLC Therapies

### 5.1. Integration with Targeted Agents

Pharmacological blockade of LAT1 with JPH203 suppresses mTORC1 signaling [220]. In preclinical NSCLC models, LAT1 inhibition enhanced the effect of the EGFR-TKI gefitinib when using the competitive substrate BCH [106]. This was combined with JPH203-mediated radiosensitization, rather than EGFR-TKI synergy [182], suggesting a rational combination strategy with transporter-targeted α-therapy. However, resistance to kinase inhibitors inevitably emerges, typically within 8–14 months for first- and second-generation EGFR TKIs [221] and through diverse adaptations to KRASG12C inhibitors [222]. In this context, transporter-targeted ^211^At ligands offer a mechanistically orthogonal approach [223], as α-particles induce clustered DNA double-strand breaks that are poorly repaired [224]. Their cytotoxicity is largely oxygen-independent (OER ≈ 1), enabling efficacy in hypoxic niches where resistant clones persist [139]. Preclinical studies with LAT1-directed ligands such as ^211^At-AAMT confirm robust tumor uptake and suppression in xenograft models [63]. Collectively, these features support the rationale for exploring transporter-targeted ^211^At conjugates as complementary partners to EGFR and KRAS inhibitors, particularly for compound-resistant mutations with limited treatment options.

### 5.2. Therapeutic Synergy with Radiotherapy and Chemotherapy

Radiotherapy remains a central component of NSCLC management, particularly in cases of locally advanced disease [225]. Its efficacy, however, is constrained in hypoxic tumors, where the OER for low-LET photons approaches 2.5–3 [77]. By contrast, high-LET α-particles show markedly reduced oxygen dependence (with OER values approaching unity), enabling effective killing in hypoxic niches [139]. Targeting amino acid transporters further amplifies this synergy: in KEAP1/NRF2-activated contexts, SLC7A11 (xCT) dependency can be exploited [161], pharmacologic xCT inhibition with sulfasalazine depletes glutathione, elevates ROS, and enhances cisplatin cytotoxicity in preclinical models [226]. LAT1 inhibition (e.g., JPH203) sensitizes NSCLC cells to irradiation by downregulating mTOR signaling and promoting radiation-induced senescence [182]. Taken together, these mechanisms—reduced hypoxia sensitivity with α-radiation, heightened redox stress, and altered damage responses—provide a mechanistic rationale for integrating transporter-targeted α-therapy (e.g., ^211^At agents) with radiotherapy or platinum chemotherapy to broaden the therapeutic index and suppress resistant subpopulations.

### 5.3. Synergy with Immunotherapy

Metabolic interventions provide a unique bridge between α-particle therapy and immune checkpoint blockade. Inhibition of glutamine uptake with V-9302 enhances antitumor immunity in TNBC models [109], illustrating how transporter blockade remodels the tumor–immune interface. α-particle irradiation itself can stimulate immunogenic cell death (ICD), with DAMP exposure such as HMGB1 and calreticulin [227]. For example, ^223^Ra treatment increases the exposure of HMGB1 and calreticulin, promoting dendritic cell maturation and T-cell priming [228]. At the transporter–immune axis, LAT1 inhibition reduces PD-L1 expression in NSCLC cells [220], while xCT blockade disrupts glutathione homeostasis and supports combination with checkpoint blockade [229]. Preclinical evidence shows that α-therapy plus PD-L1 antibodies achieves superior tumor control compared with either modality alone [230]. Importantly, treatment sequencing remains unresolved: concurrent administration outperformed sequential regimens in peptide-based TRT models [231]. This synergy may be particularly impactful in oncogene-driven NSCLC, where ICIs alone have shown limited benefit [25,26], positioning transporter-targeted α-therapy as a mechanistically orthogonal partner to immunotherapy.

### 5.4. Synergies with Ferroptosis and DNA Damage

^211^At α-particles induce clustered, irreparable DNA double-strand breaks with minimal oxygen dependence [188,224]. When combined with the inhibition of ASCT2 and xCT, glutamine- and cystine-dependent metabolic circuits are disrupted, resulting in the depletion of NAD(P)H and glutathione, and amplifying ferroptotic cell death [100,232]. This effect is particularly relevant in KEAP1/NRF2-altered tumors, where NRF2-driven antioxidant programs confer resistance to both ferroptosis and radiation [233]. Parallel inhibition of mTOR signaling with LAT1 blockade (e.g., JPH203) further sensitizes tumors to irradiation [182]. These effects are mechanistically linked to the transporter-mediated metabolic adaptations outlined in Section 3.2, highlighting the potential of integrated targeting strategies.

### 5.5. Theranostic Framework

^211^At-labeled transporter ligands could serve within a unified theranostic framework that links patient selection, individualized treatment planning, and longitudinal monitoring [234]. LAT1-targeted PET tracers (e.g., ^18^F-FAMT, ^18^F-FBPA) allow noninvasive assessment of LAT1 activity and can aid treatment planning (e.g., BNCT selection using quantitative uptake) [58]. Similarly, [^18^F]FSPG PET captures xCT/NRF2 activity, and heterogeneous retention in NSCLC tumors offers a complementary biomarker for therapeutic stratification [112,235]. Preclinical PET studies further reveal that intra- and inter-tumoral uptake heterogeneity often contrasts with homogeneous ex vivo receptor expression, underscoring the predictive value of functional imaging for therapies requiring uniform target engagement [236]. Clinically, for NSCLC with KEAP1/NFE2L2 mutations, the benefit from immune checkpoint inhibitors remains uncertain, with conflicting data [237]. In EGFR-mutant resistance models, NRF2 inhibition restores vulnerability and supports a rationale for combinations with EGFR-TKIs [238]. These insights highlight transporter-based imaging as a companion biomarker to guide adaptive trial designs that integrate imaging–dosimetry feedback. Lessons from early-phase α-therapy trials, such as ^225^Ac-PSMA in prostate cancer, confirm the feasibility of this paradigm [211]. Collectively, transporter-targeted ^211^At theranostics may evolve from experimental probes into integral components of multimodal NSCLC management, with priorities including the development of standardized biomarkers, dosing algorithms, and the conduct of harmonized multi-center trials.

## 6. Conclusions and Future Perspectives

NSCLC remains a significant cause of cancer-related mortality, mainly because therapeutic resistance persists despite multimodal advances. Amino acid transporters—particularly LAT1, ASCT2, and xCT—serve as critical regulators of tumor metabolism and immune evasion, offering rational entry points for therapeutic intervention and molecular imaging. Among α-emitters, ^211^At combines an optimal half-life, high LET, and short-range energy deposition, which enable precise, mechanism-driven targeting. By exploiting these properties, ^211^At-labeled ligands directed at amino acid transporters provide an orthogonal strategy to overcome resistance and intratumoral heterogeneity—two enduring barriers to durable response in NSCLC. These advantages define the rationale for integrating ^211^At into translational frameworks that link radiochemistry, tumor metabolism, and clinical theranostics.

Looking ahead, critical priorities include biomarker-driven patient selection—for example, LAT1- and xCT-targeted PET imaging and multi-omics profiling; optimization of delivery platforms, including small molecules, antibodies, peptides, and nanocarriers; and integration into adaptive theranostic frameworks that enable longitudinal monitoring and individualized dosing. Rational combinations with EGFR and KRAS inhibitors, immunotherapy, radiotherapy, and ferroptosis inducers will be essential to maximize therapeutic efficacy. Early-phase clinical translation should leverage lessons from ^223^Ra and ^225^Ac α-therapy trials, supported by GMP-compliant automation and international cooperative networks.

With continued progress in radiochemistry, tumor biology, and translational infrastructure, ^211^At-based transporter targeting is expected to evolve from experimental innovation toward clinical implementation, representing a promising component of next-generation multimodal NSCLC therapy that may ultimately yield durable survival benefits for patients worldwide.

## Figures and Tables

**Figure 1 ijms-26-10736-f001:**
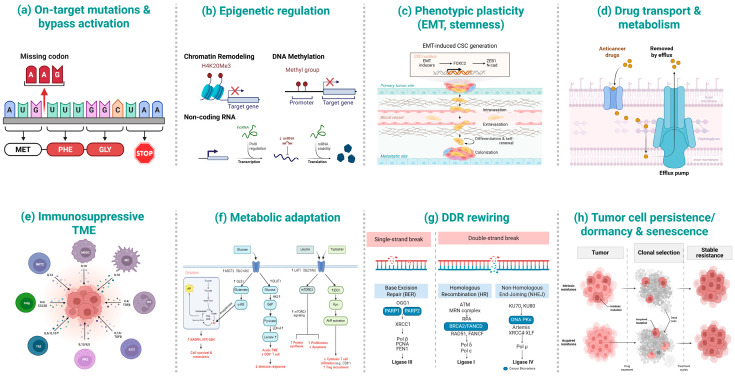
Schematic overview of representative resistance mechanisms in NSCLC. (**a**) On-target mutations & bypass activation; (**b**) Epigenetic regulation (chromatin remodeling, DNA methylation, non-coding RNAs); (**c**) Phenotypic plasticity (EMT, stemness); (**d**) Drug transport & metabolism (efflux pumps and drug handling); (**e**) Immunosuppressive tumor microenvironment (TME) (myeloid, lymphoid and stromal components around the tumor); (**f**) Metabolic adaptation (LAT1/ASCT2/xCT-centered nutrient and redox pathways); (**g**) DNA-damage response (DDR) rewiring (BER, HR, NHEJ and checkpoint signaling); (**h**) Tumor cell persistence/dormancy & senescence (clonal selection and stable resistance).

**Figure 2 ijms-26-10736-f002:**
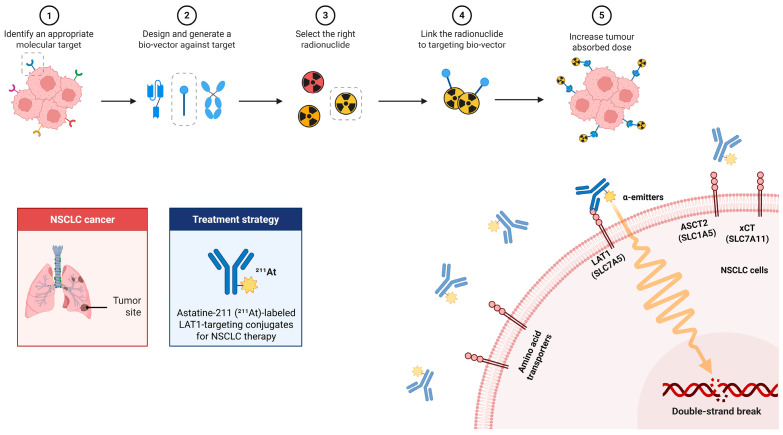
Workflow and mechanism of ^211^At-labeled LAT1-targeted therapy in NSCLC. Workflow (top row, steps 1–5): (1) Identify a molecular target (LAT1 in NSCLC). (2) Design a targeting bio-vector (e.g., antibody/scFv, peptide, small molecule); gray dashed icons indicate alternative formats not selected. (3) Select the radionuclide (preferred α-emitter ^211^At; other candidates shown as alternatives). (4) Link the radionuclide to the bio-vector to form the targeting conjugate. (5) Increase tumor absorbed dose via specific uptake at the tumor site. Mechanism (bottom right): LAT1 (SLC7A5) on the NSCLC-cell membrane mediates targeting of the ^211^At-labeled conjugate; emitted α-particles (wavy track) deposit high-LET energy over a short range, causing DNA double-strand breaks. ASCT2 (SLC1A5) and xCT (SLC7A11) are shown as related transporters for contextual comparison.

**Table 1 ijms-26-10736-t001:** Representative resistance mechanisms in NSCLC, with key molecular alterations, functional consequences, and therapeutic implications.

Mechanism	Key Alterations/Molecules	Functional Consequences	Clinical/Therapeutic Relevance
On-target mutations & bypass activation	EGFR T790M/C797S [34]; ALK G1202R [35]; MET amplification [36]	Impaired drug binding [34,35]; bypass reactivation of PI3K/AKT and MAPK signaling [36]	Supports next-generation TKIs and combinatorial regimens (e.g., osimertinib + MET inhibitors [37])
Epigenetic regulation	DNMT1/3A [38,39]; EZH2 [40]; HDACs [41]; miR-21 [42]	TSG silencing [38,39]; drug-tolerant state maintenance [40]	Provides rationale for DNMT/HDAC/EZH2 inhibitors and epigenetic–TKI/ICI combinations [43]
Phenotypic plasticity (EMT, stemness)	EMT-TFs [44], YAP/TAZ [45]	Promotes migration, invasion, and drug tolerance [46]	Highlights EMT/CSC-targeted and microenvironmental interventions [47]
Drug transport & metabolism	ABCB1/ABCG2 (efflux) [48]; CYP/UGT variants [49,50]	Reduced intracellular drug levels [48]; altered clearance/exposure [49,50]	Informs transporter-sparing drug design [51], nanoparticle delivery [52], and PK-guided dosing [53]
Immunosuppressive TME	PD-L1 upregulation [54]; Treg/MDSC infiltration [55,56]	CD8^+^ T-cell exclusion and impaired antigen presentation [57]	Supports next-generation ICIs (e.g., PD-1 + LAG-3) and myeloid/Treg-targeted strategies [54]
Metabolic adaptation	LAT1 [58], ASCT2 [59], xCT [60]	Enhanced glutamine/cystine influx → redox control, biosynthesis, immune modulation [60,61,62]	Creates vulnerabilities for metabolic inhibitors and radiolabeled strategies (e.g., LAT1-directed α-therapy [63]
DNA damage response (DDR) rewiring	HR/NHEJ modulation [64]; ATR–CHK1–WEE1 axis [65]	Enhanced repair and replication-stress tolerance [66]	Provides rationale for DDR inhibitors combined with chemo/radiotherapy or TKIs [67,68]
Tumor cell persistence/dormancy & senescence	Drug-tolerant persister (DTP) states [69]; therapy-induced senescence (SASP) [70]	Minimal-residual disease survival and relapse risk [69,71]	Supports persister eradication and senolytic strategies to prevent recurrence [72]

**Table 2 ijms-26-10736-t002:** Amino acid transporters in NSCLC, with structural characteristics, transport substrates, functional roles, and clinical relevance.

Transporter	Structural Features	Functional Role	Clinical Relevance
LAT1 (SLC7A5)	Disulfide-linked heterodimer with CD98hc/4F2hc (SLC3A2) [93]	Leucine uptake → mTORC1 activation [94]	Overexpressed/prognostic in NSCLC; therapeutic target [94,95]
ASCT2 (SLC1A5)	Na^+^-dependent obligatory antiporter [96]; N-glycosylation at N163/N212 [85]	Glutamine uptake sustaining NSCLC proliferation [97]	Correlates with stage, lymphatic/vascular invasion; independent poor-prognosis marker (NSCLC/AC) [98]
xCT (SLC7A11)	Light chain of system X_c_^−^; heterodimer with 4F2hc [99]	Cystine/Glutamate exchange → GSH synthesis/antioxidant defense [100]	Upregulated in NSCLC; linked to poor prognosis and drug resistance [100]; therapeutic target [101]

**Table 3 ijms-26-10736-t003:** Therapeutic strategies targeting amino acid transporters and ^211^At radiopharmaceuticals in NSCLC.

Therapeutic Strategy	Representative Agents	Mechanism of Action	Evidence/Preclinical Findings
LAT1 (SLC7A5) inhibitors	JPH203 (nanvuranlat) [182], BCH [106], benzylserine [183]	Block leucine influx via LAT1 → dampen mTORC1 [182]; cytostatic G0/G1 arrest [183]	JPH203 radiosensitizes cancer cells via mTOR downregulation [182]; BenSer inhibits Leu/Gln uptake and cell cycle progression [183].
ASCT2 (SLC1A5) inhibitors	V-9302 [158], benzylserine [183]	Reduce glutamine influx, limiting anaplerosis [184]; condition TME for ICI (*note:* V-9302 target selectivity debated [159])	V-9302 attenuates tumor growth [184]; glutamine blockade enhances the efficacy of checkpoint blockade in lung cancer models [185].
xCT (SLC7A11) inhibitors	Erastin [164], sulfasalazine [161], sorafenib [164]	Block cystine uptake → deplete GSH → induce ferroptosis [163]	Erastin sensitizes xCT^+^ tumors to ionizing radiation [162]; SSZ limits cystine uptake via xCT, thereby lowering GSH and increasing oxidative stress, resulting in growth inhibition [100].
Monoclonal antibodies & ADCs	Anti-LAT1 mAbs (SOL22, SOL69) [186]; IGN523 (anti-CD98hc) [171]; CD98hc-ADCs [173]	mAb internalization → AA uptake decrease ↓; ADCC [187]; ADC payload delivery [173]	Anti-LAT1 mAbs: internalization/ADCC and in vivo antitumor effects [184]; IGN523: lymphoma xenograft activity; AML clinically evaluated; CD98 broadly expressed, incl. NSCLC [171]; CD98hc-ADC active in vivo [173].
^211^At-LAT1 radioligands	^211^At-AAMT [188]; ^211^At-AAMP [155]	LAT1-mediated uptake delivers α-tracks, causing clustered DSBs [63]	^211^At-AAMT: high LAT1 affinity, induces DSBs [63]; ^211^At-AAMP: growth inhibition and survival benefit in vivo [155].
Pretargeted strategies (PRIT)	^211^At-tetrazine [189] + inverse electron-demand Diels–Alder (IEDDA) [189]	Decouple carrier/radionuclide → improve tumor-to-blood ratios [189]	^211^At-labeled pretargeting agent shows higher tumor-to-blood ratios vs. directly labeled mAbs [186]; IEDDA PRIT supports α-therapy [190].
Combination approaches	LAT1 inhibitors + RT/mTOR inhibitors [116]; xCT inhibitors [82] + RT/FINs	Synthetic lethality via DNA damage + metabolic stress; radiosensitization [182]	JPH203 + RT radiosensitization [182]; xCT inhibition enhances RT response [162]; glutamine blockade augments ICI efficacy [185].

## Data Availability

The original contributions presented in this study are included in the article. Further inquiries can be directed to the corresponding author(s).

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
