# Peer review of "Astatine-211-Labeled Therapy Targeting Amino Acid Transporters: Overcoming Drug Resistance in Non-Small Cell Lung Cancer"

_ijms, 2025, doi:10.3390/ijms262110736_

Round 1
Reviewer 1 Report
Comments and Suggestions for Authors
The physical properties that need to be introduced in the Introduction;
Too much Keywords ;
It is necessary to introduce the drug resistance of ²¹¹At in other cancer applications;
It is necessary to introduce the application of Astatine-211-Labeled Therapy Targeting Amino Acid Transporters in other cancers;
There are also grammatical errors that need to be carefully checked;
The format of references should be uniform.
Author Response
Comments 1: The physical properties that need to be introduced in the Introduction;
Response 1: Thank you for this valuable suggestion.
We have revised the Introduction to include a concise summary of the radiophysical and radiochemical characteristics of astatine-211 (²¹¹At) and α-particle therapy (Page 2, Lines 52–59).
Revised version: Targeted α-particle therapy deposits large amounts of energy within a very short path length, generating densely ionizing tracks that induce complex and primarily irreparable DNA double-strand breaks, while maintaining cytotoxicity largely independent of oxygen tension [12]. Astatine-211 (²¹¹At) exhibits favorable radiophysical and radiochemical characteristics—including high linear energy transfer, short tissue penetration range, and a simple decay scheme yielding one α-particle and ending in stable ²⁰⁷Pb—that collectively support reliable radiopharmaceutical design and centralized clinical distribution [13].
Reason for revision: The previous version mentioned α-particle therapy but did not explain why ²¹¹At is particularly suitable for targeted radiotherapy. We therefore (i) explicitly stated the high-LET, micrometer-scale range, and low oxygen-dependence of α-particles, and (ii) corrected the ²¹¹At decay description from a “simple single-branch” phrasing to a dual-branch scheme that still yields one α per parent decay and ultimately results in stable ²⁰⁷Pb, with no long-lived α-emitting daughters, while refining wording on supply to “centralized manufacturing with regional distribution.”
Effect of revision: This addition clearly introduces the fundamental properties that distinguish ²¹¹At among α-emitters, thereby clarifying its radiobiological rationale and strengthening the scientific foundation of the review.
Comments 2: Too much Keywords ;
Response 2: Thank you for this valuable suggestion.
We have refined the Keywords section to reduce redundancy and improve precision while maintaining complete alignment with the manuscript’s scientific scope (Page 1, Lines 36–38).
Revised version:
Keywords: Non-small cell lung cancer; Drug resistance; Amino acid transporters (LAT1, ASCT2, xCT); Metabolic reprogramming; Astatine-211; Targeted α-particle therapy; Radiopharmaceuticals
Reason for revision: The original list contained overlapping and partly subordinate terms. To improve clarity, we consolidated related items under a single comprehensive entry—Amino acid transporters (LAT1, ASCT2, xCT)—and removed the standalone keyword “Tumor metabolism,” replacing it with the more precise term “Metabolic reprogramming,” which better reflects the mechanistic focus discussed in the manuscript.
Effect of revision: This refinement reduces the number of keywords from ten to seven, avoids redundancy, and provides a clearer hierarchy from disease(NSCLC) → mechanism (drug resistance, metabolic reprogramming) → molecular targets (amino acid transporters) → therapeutic strategy (²¹¹At, targeted α-particle therapy, radiopharmaceuticals). These changes make the keywords concise yet comprehensive, fully representing the paper’s central themes. Importantly, no new concepts or citations were introduced; existing terms were reorganized for clarity.
Comments 3: It is necessary to introduce the drug resistance of ²¹¹At in other cancer applications;
Response 3: Thank you for this valuable suggestion.
In response, we have added a new paragraph discussing mechanistic determinants of therapeutic resistance and translational limitations of ²¹¹At-based transporter-targeted α-therapy across other cancer types. The new content has been incorporated into Section 4.7, “Toward Clinical Translation” (Page 11, Lines 393–405), immediately before subsection 4.7.1, Dosimetry and Pharmacokinetics. Please refer to the detailed edits shown in the review mode.
Newly added text: Across α-therapy applications beyond NSCLC, variability in response depth and durability indicates recurrent constraints relevant to ²¹¹At transporter-targeted strategies. Intratumoral heterogeneity of radiopharmaceutical uptake [199] and spatially uneven target/transporter expression [200] can generate micro-dosimetric cold spots, compromising tumor control. Meanwhile, activation of ATM/BRCA1-linked DNA-damage responses can mitigate radiation cytotoxicity [201] and regional hypoxia/perfusion deficits [202], further limiting the effectiveness of α-particle lethality and dose deposition. Metabolic compensation via ASCT2/xCT may restore amino-acid flux when glutamine transport is perturbed [203], and pharmacokinetic liabilities—including in vivo deactivation and short half-life–related constraints—reduce target exposure [204]. Within this context, optimization of dosimetry and pharmacokinetics, structured mitigation of hematologic and thyroidal toxicity, and biomarker-guided selection using transporter PET, together with GMP-ready automation, are central considerations for advancing ²¹¹At in NSCLC.
Reason for revision: The original manuscript focused primarily on the therapeutic potential of ²¹¹At-labeled transporter ligands in NSCLC. However, as the reviewer correctly noted, drug resistance and variable therapeutic response have also been reported in other malignancies, providing valuable translational insights. To address this point, the newly added paragraph summarizes common resistance-related factors observed across α-particle therapies beyond NSCLC, including:
• Intratumoral heterogeneity of radiopharmaceutical uptake and LAT1 expression, generating microdosimetric “cold spots” that reduce tumor control;
• Activation of DNA-damage–response pathways (ATM/BRCA1) that mitigate α-particle lethality;
• Hypoxia and perfusion deficits limiting dose deposition;
• Metabolic compensation through ASCT2/xCT upregulation restoring amino-acid flux after LAT1 blockade;
• Pharmacokinetic liabilities such as rapid systemic clearance or in-vivo deactivation of small-molecule vectors.
Rationale for placement: This paragraph was added under Section 4.7 (Toward Clinical Translation) because this section transitions from preclinical radiochemistry and biology (Sections 4.1–4.6) to clinical feasibility. Placing the discussion here provides a natural bridge from mechanistic understanding to translational challenges—emphasizing that addressing resistance mechanisms is an integral part of advancing ²¹¹At toward human trials. By situating this content immediately before 4.7.1 (Dosimetry and Pharmacokinetics), we connect mechanistic resistance constraints to subsequent solutions such as dosimetry optimization, toxicity mitigation, and biomarker-guided patient selection.
Effect of the revision: This addition expands the review’s translational depth by integrating resistance phenomena observed in other tumor types, highlighting that therapeutic variability in ²¹¹At applications arises from both biological (heterogeneity, repair, metabolism) and pharmacologic (clearance, delivery) factors. It also clarifies how overcoming these barriers—through improved pharmacokinetics, GMP-compliant automation, and PET-based patient stratification—is essential for clinical advancement in NSCLC.
Comments 4: It is necessary to introduce the application of Astatine-211-Labeled Therapy Targeting Amino Acid Transporters in other cancers;
Response 4: Thank you for this valuable suggestion.
In response, we have added a new subsection titled “4.5. Applications in Other Malignancies”, immediately following 4.4. Targeting Amino Acid Transporters with ²¹¹At.
Please refer to the changes shown in review mode for the newly added content and the corresponding section renumbering.
Summary of structural changes:
- New 4.5: Applications in Other Malignancies (Page 9, Lines 318–339).
- The former 4.5. Delivery Platforms for Enhanced Retention is now 4.6, with its subitems 4.6.1–4.6.3 (Antibodies/ADCs; Peptides; Nanocarriers) renumbered accordingly.
- The former 4.6. Toward Clinical Translation is now 4.7, with its subitems 4.7.1–4.7.3 (Dosimetry and Pharmacokinetics; Safety; Patient Selection and Trial Design) renumbered in sequence.
- No changes were made to Sections 1–3 or Section 5.
Reason for adding this subsection: The original manuscript primarily focused on NSCLC and the mechanistic basis of transporter-targeted ²¹¹At therapy. To address the reviewer’s request for broader applicability, the new 4.5 summarizes representative preclinical studies in pancreatic cancer, melanoma, ovarian cancer, glioma, and sarcoma, emphasizing transporter-mediated uptake, dose-dependent efficacy, and favorable tolerability across tumor types. This addition demonstrates that the therapeutic principle is not exclusive to NSCLC, thereby reinforcing the translational foundation of the review.
Rationale for placement: Chapter 4 develops the logical sequence from principles to practice:
- 1–4.3: physical and radiochemical foundations of ²¹¹At and its labelling strategies;
- 4: targeting of amino acid transporters with ²¹¹At;
- → New 4.5: cross-cancer validation of these principles, highlighting external generalizability;
- 6–4.7: delivery platforms and clinical translation;
Section 5: integration of validated concepts back into NSCLC-specific therapeutic frameworks.
This progression—mechanistic rationale → cross-cancer validation → translational engineering → NSCLC integration—enhances structural coherence and avoids redundant general discussion in Section 5.
Effect of the revision: The addition of Section 4.5 broadens the translational scope without diluting the NSCLC focus, demonstrating that ²¹¹At-labeled transporter-targeted strategies are grounded in mechanism-driven biology validated across multiple malignancies. It also provides a smoother conceptual bridge into the NSCLC-focused integration and combination strategies discussed in Section 5, improving both coherence and readability of the manuscript.
Comments 5: There are also grammatical errors that need to be carefully checked;
Response 5: Thank you for this helpful comment.
We have carefully rechecked the entire manuscript for grammar, spelling, and stylistic accuracy. Minor grammatical and typographical errors were corrected to improve clarity, readability, and consistency with academic English standards.
Comments 6: The format of references should be uniform.
Response 6: Thank you for this valuable comment.
We have thoroughly checked and standardized all references according to the IJMS reference style, ensuring uniformity in author names, journal abbreviations, punctuation, and DOI formatting. The reference list has been fully updated for consistency and accuracy.

Reviewer 2 Report
Comments and Suggestions for Authors
- The article "Astatine-211-Labeled Therapy Targeting Amino Acid Transporters: Overcoming Drug Resistance in Non-small Cell Lung Cancer" is a review article. The title of the review article reflects its content.
- In the Abstract section, the authors briefly summarized the article. The article focuses on the search for approaches to overcoming therapy resistance in non-small cell lung cancer (NSCLC). The authors presented the key mechanisms of tumor immune evasion in NSCLC patients. They presented compelling arguments in favor of developing a mechanistic orthogonal strategy for overcoming resistance and tumor heterogeneity based on the biology of α-particle-delivered transporters and a theranostic agent based on Astatine-211.
- The "Keywords" presented in the article correspond to the content of the article and are necessary.
- In Section 1. Introduction, the authors noted that NSCLC is the leading cause of lung cancer mortality worldwide. The mechanisms of therapeutic resistance, particularly metabolic reprogramming, are presented. In this regard, the amino acid transporters LAT1 (SLC7A5) and ASCT2 (SLC1A5) are potential therapeutic targets for targeted therapy. To improve the effectiveness of such targeted therapy, the authors propose the use of targeted α-particle therapy.
- The article is presented in 6 sections: "2. Therapeutic Approaches and Mechanisms of Drug Resistance in NSCLC," "3. Amino Acid Transporters in NSCLC: Structure, Function, and Therapeutic Targeting," "4. Physical and Radiobiological Basis of 211At Therapy," and "5. Integration with Existing NSCLC Therapies." The content of the sections corresponds to their titles. Tables and figures are important for understanding the article. In section "6. Conclusion and Future Perspectives," the authors summarized their analysis of the topic. However, the data in the first and second paragraphs of section "6. Conclusion and Future Perspectives" have been repeatedly presented in other sections of this article. Frequent repetition of the same material reduces the reader's favorable attitude towards the article. In this regard, we strongly recommend that authors focus their attention on paragraphs 3, 4, and 5 of Section 6. This will highlight the novelty of the article: a translational framework for overcoming therapeutic resistance in NSCLC is proposed.
- The article is interesting and important for clinical practice. The text is clearly written. The manuscript did not raise any ethical concerns. All references in the "References" section are formatted correctly.
- After the authors' revisions to the article, I will have no concerns about its similarity to other articles published by the same authors.
- The authors' competing interests do not bias the presentation of their results and conclusions.
Author Response
Comments 1: The article "Astatine-211-Labeled Therapy Targeting Amino Acid Transporters: Overcoming Drug Resistance in Non-small Cell Lung Cancer" is a review article. The title of the review article reflects its content.
Response 1: Thank you for this positive comment.
We appreciate the reviewer’s recognition that the title accurately reflects the content and scope of the manuscript. No changes were made to the title.
Comments 2: In the Abstract section, the authors briefly summarized the article. The article focuses on the search for approaches to overcoming therapy resistance in non-small cell lung cancer (NSCLC). The authors presented the key mechanisms of tumor immune evasion in NSCLC patients. They presented compelling arguments in favor of developing a mechanistic orthogonal strategy for overcoming resistance and tumor heterogeneity based on the biology of α-particle-delivered transporters and a theranostic agent based on Astatine-211.
Response 2: Thank you for this positive and encouraging comment.
We carefully reviewed the Abstract and made minor terminology refinements to improve precision and ensure consistency with the revised Introduction (Page 1, Lines 16–35). All edits are visible in track changes; no new concepts were added and the original meaning remains unchanged.
Reason for revision: To harmonize wording and formatting with the main text, we standardized several expressions and notations. Specifically, we replaced “major” with “primary” and “alterations” with “changes,” aligned the mechanistic phrasing to “metabolic reprogramming driven by amino-acid transporters” with the same transporter listing as in the body, and normalized verbs/nouns such as “supports”→“sustains,” “proliferation”→“growth,” “homeostasis”→“balance,” and “escape”→“evasion.” The α-therapy sentence was unified to “has emerged as a promising modality due to its capacity to eradicate resistant clones via clustered, irreparable DNA double-strand breaks,” we adopted the consistent nuclide wording “predictable decay scheme,” used “demonstrated” for preclinical evidence and “ferroptosis induction” for pathway terminology, and normalized PET tracer notation to [18F]FAMT/[18F]FSPG. We also removed a redundant “further” in “collectively support,” kept “theranostic framework” and “²¹¹At-based theranostics” consistent, and corrected the capitalization/duplication to “Successful translation.”
Effect of revision: The Abstract now reads more fluently and uses standardized scientific terminology consistent with the rest of the manuscript, improving clarity and readability for an international audience while fully retaining the original scope and conclusions.
Comments 3: The "Keywords" presented in the article correspond to the content of the article and are necessary.
Response 3: Thank you for this positive and encouraging comment.
Acknowledging the reviewer’s assessment, we made only a light structural refinement to enhance readability and thematic organization while preserving all essential concepts (Page 1, Lines 36–38).
Revised version:
Keywords: Non-small cell lung cancer; Drug resistance; Amino acid transporters (LAT1, ASCT2, xCT); Metabolic reprogramming; Astatine-211; Targeted α-particle therapy; Radiopharmaceuticals
Reason for revision: Rather than changing the scientific scope, we optimized the presentation: closely related transporter terms were grouped under “Amino acid transporters” (LAT1, ASCT2, xCT), and the broad term “Tumor metabolism” was removed and replaced with “Metabolic reprogramming” to align with the manuscript’s mechanistic emphasis and improve indexing specificity.
Effect of revision: The revised list maintains every key concept noted by the reviewer while improving logical flow from disease → mechanism → targets→ therapeutic strategy. The refinement enhances clarity without introducing new concepts or references; it simply reorganizes existing terms and clarifies the mechanistic wording by substituting 'Metabolic reprogramming' for the former 'Tumor metabolism'.
Comments 4: In Section 1. Introduction, the authors noted that NSCLC is the leading cause of lung cancer mortality worldwide. The mechanisms of therapeutic resistance, particularly metabolic reprogramming, are presented. In this regard, the amino acid transporters LAT1 (SLC7A5) and ASCT2 (SLC1A5) are potential therapeutic targets for targeted therapy. To improve the effectiveness of such targeted therapy, the authors propose the use of targeted α-particle therapy.
Response 4: We appreciate the reviewer’s positive assessment.
To strengthen the logical transition from NSCLC resistance mechanisms and amino-acid transporter targeting to α-particle therapy, we refined the paragraph describing α-particle properties (Page 2, Lines 52–59).
Revised version: Targeted α-particle therapy deposits large amounts of energy within a very short path length, generating densely ionizing tracks that induce complex and primarily irreparable DNA double-strand breaks, while maintaining cytotoxicity largely independent of oxygen tension [12]. Astatine-211 (²¹¹At) exhibits favorable radiophysical and radiochemical characteristics—including high linear energy transfer, short tissue penetration range, and a simple decay scheme yielding one α-particle and ending in stable ²⁰⁷Pb—that collectively support reliable radiopharmaceutical design and centralized clinical distribution [13].
Reason for revision: The previous text summarized α-particle therapy but did not specify why ²¹¹At is particularly suitable. We therefore clarified the biophysical basis (high LET, micrometer-scale range, oxygen-tension independence) and corrected the decay description from “single-step decay to stable ²⁰⁷Pb” to “a simple decay scheme yielding one α-particle and ending in stable ²⁰⁷Pb,” aligning with the cited references and the tracked changes in the manuscript.
Effect of revision: These refinements provide the necessary radiophysical context and create a smoother linkage from transporter-based resistance mechanisms to α-particle therapy, improving clarity and coherence in the Introduction without altering the manuscript’s conclusions.
Comments 5: The article is presented in 6 sections: "2. Therapeutic Approaches and Mechanisms of Drug Resistance in NSCLC," "3. Amino Acid Transporters in NSCLC: Structure, Function, and Therapeutic Targeting," "4. Physical and Radiobiological Basis of 211At Therapy," and "5. Integration with Existing NSCLC Therapies." The content of the sections corresponds to their titles. Tables and figures are important for understanding the article. In section "6. Conclusion and Future Perspectives," the authors summarized their analysis of the topic. However, the data in the first and second paragraphs of section "6. Conclusion and Future Perspectives" have been repeatedly presented in other sections of this article. Frequent repetition of the same material reduces the reader's favorable attitude towards the article. In this regard, we strongly recommend that authors focus their attention on paragraphs 3, 4, and 5 of Section 6. This will highlight the novelty of the article: a translational framework for overcoming therapeutic resistance in NSCLC is proposed.
Response 5: Thank you for this valuable and constructive suggestion.
In the revised manuscript, the first two paragraphs of Section 6 (Conclusion and Future Perspectives) were merged and rewritten to eliminate redundant mechanistic and radiochemical descriptions that were already discussed in Sections 2–4 (Page 14, Lines 543–550; Page 15, Lines 551–562). Please refer to the changes shown in the review mode for detailed edits.
Reason for revision: The revision aimed to improve logical flow and avoid redundancy, while maintaining a concise overview of the therapeutic resistance challenge, the rationale for targeting amino acid transporters, and the distinctive properties of ²¹¹At.
Effect of revision: The new version transitions more smoothly toward the translational framework that links radiochemistry, tumor metabolism, and clinical theranostics.
Paragraphs 3–5 were retained and refined to emphasize biomarker-based patient selection, individualized dosing, and rational combination strategies.
These changes eliminate repetition and clearly highlight the novelty of the review—its mechanistically grounded translational approach to overcoming therapeutic resistance in NSCLC.
Comments 6: The article is interesting and important for clinical practice. The text is clearly written. The manuscript did not raise any ethical concerns. All references in the "References" section are formatted correctly.
Response 6: Thank you for these positive and encouraging comments.
We are grateful for the reviewer’s recognition of the manuscript’s clinical relevance, clarity, and adherence to ethical and formatting standards. No changes were required in this regard.
Comments 7: After the authors' revisions to the article, I will have no concerns about its similarity to other articles published by the same authors.
Response 7: Thank you for this positive comment.
We appreciate the reviewer’s acknowledgment and confirmation that the revised manuscript is clearly distinct from our previous publications. We confirm that this review is original, has not been published elsewhere, and does not overlap with any of our prior work.
Comments 8: The authors' competing interests do not bias the presentation of their results and conclusions.
Response 8: Thank you for this positive comment.
We appreciate the reviewer’s acknowledgment of the objectivity of our work. We confirm that there are no competing interests influencing the results or conclusions, and no changes were required in this regard.
